# The Impact of Air Pollution on Gut Microbiota and Children’s Health: An Expert Consensus

**DOI:** 10.3390/children9060765

**Published:** 2022-05-24

**Authors:** Eddy Fadlyana, Dewi Sumaryani Soemarko, Anang Endaryanto, Budi Haryanto, Andy Darma, Dian Kusuma Dewi, Dian Novita Chandra, Budi Hartono, Sonia Buftheim, Erika Wasito, Tonny Sundjaya, Ray Wagiu Basrowi

**Affiliations:** 1Department of Growth and Development-Social Pediatrics, Faculty of Medicine, Universitas Padjadjaran, Bandung 45363, Indonesia; edfadlyana@yahoo.com; 2Department of Community Medicine, Faculty of Medicine, Universitas Indonesia, Jakarta 10430, Indonesia; dewisoemarko@yahoo.com (D.S.S.); dian.kusuma.dewi@gmail.com (D.K.D.); 3Department of Child Health, Faculty of Medicine, Universitas Airlangga, Surabaya 60132, Indonesia; aendaryanto.ae@gmail.com (A.E.); andy.darma@fk.unair.ac.id (A.D.); 4Department of Environmental Health, Faculty of Public Health, Universitas Indonesia, Jakarta 10430, Indonesia; bharyanto@ui.ac.id (B.H.); ngajitrus@yahoo.com (B.H.); 5Research Center for Climate Change, Institute for Sustainable Earth & Resources, Universitas Indonesia, Depok 16424, Indonesia; 6Department of Nutrition, Faculty of Medicine, Universitas Indonesia, Jakarta 10430, Indonesia; diannovitach@yahoo.com; 7Nano Interfacial Chemistry Lab-Department of Chemistry (NIC), Universitas of Indonesia, Depok 16424, Indonesia; soniaumarbuftheim@gmail.com; 8Medical and Science Affairs Division, Danone Specialized Nutrition, Jakarta 12950, Indonesia; erika.wasito@danone.com (E.W.); tonny.sundjaya@danone.com (T.S.)

**Keywords:** air pollution, children’s health, expert consensus

## Abstract

Air pollution is an unseen threat to children’s health because it may increase the risk of respiratory infection, atopy, and asthma, and also alter gut microbiota compositions. The impact of air pollution on children’s health has not been firmly established. A literature review followed by a series of discussions among experts were performed to develop a theoretical framework on how air pollution could affect various bodily organs and functions in children. We invited experts from different backgrounds, such as paediatricians, nutritionists, environmental health experts, and occupational health experts, to provide their views on this matter. This report summarizes the discussion of multidisciplinary experts on the impact of air pollution on children’s health. The report begins with a review of air pollution’s impact on allergy and immunology, neurodevelopment, and cardiometabolic risks, and ends with the conceptualization of a theoretical framework. While the allergic and immunological pathway is one of the most significant pathways for air pollution affecting children’s health in which microbiotas also play a role, several pathways have been proposed regarding the ability to affect neurodevelopment and cardiometabolic risk. Further research is required to confirm the link between air pollution and the gut microbiota pathway.

## 1. Introduction

Air pollution is one of the most significant health problems in the world; data from the World Health Organization (WHO) show that 9 out of 10 people breathe polluted air, and has been estimated that 7 million people die annually because of the combination of ambient and household air pollution [1]. However, this exposure to pollutants uniquely affects children because of a combination of behavioural, environmental, and physiological factors, thus making children more vulnerable to air pollution. Children have longer life expectancies, and therefore, longer exposure, and they breathe faster than adults, taking in more pollutants through the air. Children also live closer to the ground and may spend more time outside, exposing them to higher levels of air pollution. Physiologically, their bodies are rapidly changing and maturing, and therefore, are more vulnerable to damage from pollutants. Moreover, their powerlessness to change their environments is the reason that adults must be responsible to protect them from the threat of unsafe air [2].

While it is well-known that air pollution affects health, especially in small children, the exact underlying biomechanism is still under investigation. As has been shown, air pollution has been responsible for mortality and morbidity due to respiratory infection among children under 5 years of age. In fact, 28.9% of the disability adjusted life years (DALYs) lost to air pollution was due to respiratory infections [3]. In addition to its direct effect on the respiratory tract, which is the place where particulate matter from air pollution will enter the human body, one of the proposed mechanisms among the complex is through an imbalanced condition in the gut microbiota community, or so-called “dysbiosis”. A comprehensive literature review has shown that air pollution alters the composition of the gut microbiome, albeit mildly, but there has been little consistency among studies regarding these changes [4]. Although human studies are still limited, animal studies and observational studies have shown that air pollution is associated with the increased permeability and inflammation of the gastrointestinal tract and therefore encourages imbalance in the microbiota. The strong link between gut microbiota and the immune system and inflammation, as well as energy extraction from indigestible fibres, suggest that they may play roles in how a gut-microbiota imbalance may lead to childhood diseases, including immune diseases such as atopy, metabolic disorders such as obesity, and neurodevelopmental disorders [5].

Air pollution is a significant problem in urban Indonesia. A 2020 study conducted in 16 large, Indonesian cities showed that the majority of the average annual PM_2.5_ exceeded the Indonesian annual ambient air quality standard (15 µg m^−3^), owing mostly to traffic emissions and biomass burning. Volcanic emissions, as well as forest and peat fires, also affect some of the cities in Indonesia [6]. However, there has been no agreement yet among clinicians and researchers in Indonesia regarding how the air pollution in the urban Indonesian context affects children’s health, especially through microbiota dysbiosis. Therefore, an expert meeting involving multidisciplinary parties has been needed to contextualize the risk of air pollution in affecting children’s health, specifically through microbiota imbalances and gut permeability. By understanding how the balance of the microbiota community can affect children’s health, it is expected that some form of nurture, such as nutritional supplements, can interact and heal the damage caused by air pollution.

## 2. Materials and Methods

This report was developed based on a literature review followed by a series of discussions with multidisciplinary experts. The literature review aimed to produce a draft of a theoretical framework based on scientific evidence. The draft was then disseminated to all of the experts before the meeting occurred so that the experts could be well-prepared.

### 2.1. Literature Review

A comprehensive search of English- and Indonesian-language publications was conducted using the PubMed database. The following terms were used for searching: “air pollution” and “child’s health”. Studies were included if they discussed the link between air pollution and its effects children’s health and excluded if they reviewed very specific particles (e.g., mercury or lead). The population to which it was applied was children under the age of 5 years old. Relevant articles were graded according to their level of evidence, and only articles mentioning specific and direct links between air pollution and children’s health, based on either human or animal studies, were taken into consideration for analysis.

### 2.2. Experts Selection

We invited experts from various fields, namely three experts from paediatrics (growth and development; allergy and immunology; and gastrohepatology), two experts in human nutrition (community and clinical nutrition), and another three experts in environmental health, environmental toxicology, and occupational medicine, respectively. The eligibility criteria for the experts was that they each had at least ten years of work experience in their respective field and were mainly practitioners or academicians.

### 2.3. Procedures

The meetings were performed online via the Zoom meeting platform. We had two sessions of discussions. The first session was conducted among the experts on environmental and occupational health so as to come to an understanding of the impact of air pollution on the social and physical health of children, while the second session focused on a discussion of how air pollution affects the risk of allergy and gastroenterological problems in children. All discussions were recorded and transcribed. The experts involved were paediatricians with subspecialities in allergy, immunology, and gastrohepatology. This combination of experts from various disciplines, such as paediatric, clinical nutrition, community nutrition, and environmental health, was deliberately designed to ensure a holistic approach to understanding air pollution as one of the important social determinants of health.

### 2.4. Data Analysis

The expert meetings were recorded, transcribed verbatim, and analysed by the authors. The analysis was performed using Microsoft Excel. We identified similarities and differences in the experts’ statements and classified and coded them accordingly. The result of the expert meetings was the clarification of the draft of the theoretical framework that had been developed based on the literature review.

## 3. Results

### 3.1. Literature Review

The search strategy resulted in 346 total articles, but after screening for relevancy and level of evidence, the literature search concluded with 31 articles. The results of the literature search are described in Figure 1.

The combination of a literature review with expert meetings resulted in a summary of the impacts of air pollution on children’s health, which will be discussed in more details below.

### 3.2. Allergy and Immunology

Children exposed to oxidants, such as ozone (O_3_) and nitrogen dioxide (NO_2_), have been found to be more prone to develop asthma, allergic rhinitis, and eczema. The oxidants target the airway’s epithelium, and combined with an impaired ability for an antioxidant response, easily damage the airway’s epithelium. However, the evidence on early exposure to PM_2.5_ and PM_10_ resulting in risks of childhood respiratory and allergic conditions was conflicting [7,8]. Tobacco smoke, however, has a clearer relationship; environmental tobacco smoke (ETS) is strongly associated with asthma symptoms by increasing the airway’s hyperresponsiveness. ETS has a direct, toxic effect on the mucosa, impairing ciliary function and local immune defences, resulting in prolonged inflammation. Nicotine affects lung branching in utero, resulting in dysanaptic lung growth, decreasing airflow, increasing resistance and mucous viscosity, and therefore affects pulmonary function [9,10]. Because of the lowered immune responses, including airway clearance, the risk for respiratory infection is also increased.

The relationship between air pollution and respiratory infection is fascinating and important, as the respiratory tract is the first line of defence against air pollution. Therefore, further review is needed in regard to the way that air pollution affects respiratory function, including infection and hypersensitivity. Air pollution increases the risk of respiratory infection by impairing immune responses in the respiratory tract. PM_2.5_ causes direct lung cell injury, affecting the bacterial clearance from the airway [11]. PM_2.5_ also reduces alveolar macrophage phagocytosis of bacteria [12]. Chronic exposure to diesel exhaust particles also directly kills cells, and therefore decreases responses towards the pathogen [13]. Air pollution also causes oxidative stress within respiratory cells, increasing susceptibility to infection by increasing viral attachment and entering cells [14]. Exposure to carbon black also changes the immune response from a Th1 to a Th2, which promotes allergic immune response. This shift promotes the exacerbation of a virus which, in this study, was respiratory syncytial virus in mice [15].

Microbiota play a huge role in this relationship, as previously discussed. Air pollutants that are ingested can alter gut microbiota compositions, creating dysbiosis by changing the gut environment. The air pollutant particles affect gut epithelial cells, both directly and after being metabolized by the gut microbiota. Both encourage reactive oxygen species (ROS) production and affect epithelial tight junctions, allowing microbial products and air pollutant particles to penetrate deeper to the gut wall. This penetration increases immunological cell interaction, causing pro-inflammatory reactions that change the gut microbiota’s composition to suit this environment. The inflammation itself and the microbiota composition change—the dysbiosis—will further increase gut permeability, creating a cycle of effects [16]. Gut microbiota dysbiosis caused by air pollution also plays a role in the increased risk of atopy and asthma. The microbiota plays a role in the regulation of immune functions by directly stimulating or producing several mediators that stimulate the production of regulatory T-cells (T-regs). T-regs are a type of T-cells that modulate the activity of the immune system. While asthma may have been more affected by the inhaled antigen and lung microbiota, gut microbiota dysbiosis has also been observed in asthmatic children, indicating the possible role of gut microbiota in systemic immune response and asthma [17,18]. The data also showed that, when the gut microbiota is altered, the susceptibility for infection, both in the gut compartment and in distant organs, such as the lungs, is increased. This is caused by the membrane component of the gut microbiota and also because metabolites produced by the microbiota can induce the interferon-mediated protection from viral infection, even in the distal respiratory system [19]. While the allergic and immunological pathway is one of the most significant pathways for air pollution to affect children’s health, in which the microbiota also plays a role, several pathways have also been identified in regard to the impact of air pollution on the alteration of the gut microbiota (Figure 2).

### 3.3. Neurodevelopment

Behavioural development is significantly affected by air pollution; one example is that exposure to NO2 and PM2.5 increases the risk for attention deficit hyperactivity disorder (ADHD). The mechanism is thought to be chronic brain inflammation, leading to the damage and diffuse loss of neural tissue. Affected areas include the prefrontal cortex, olfactory bulb, and midbrain, which is central to the development of behaviour and cognitive function [20]. The same mechanism is thought to underlie other behavioural development problems, such as Alzheimer disease pathology [21]. The biomechanism of nicotine and second-hand smoke on behavioural development is also thought to follow the same pathogenesis [9,22].

Another mechanism by which air pollution affects neurodevelopment is by affecting the gut microbiota balance [4]. Brain development is modulated by microbiota through several mechanisms: metabolic, immune, endocrine, and neural pathways. Gut microbiota, such as *Bifidobacterium infantis*, regulates central neurotransmitters, such as serotonin, by modifying its precursor, tryptophan. Tryptophan is stored in limited quantities in brain and therefore requires continuous supplementation from the digestive system. Other metabolites produced by gut microbiota that may affect brain development are short-chain fatty acids. Other than metabolic products, gut microbiota can also synthesize neurotransmitters, such as gamma-aminobutyric acid (GABA), norepinephrine, dopamine, and acetylcholine; how they affect the brain directly is debatable because of their limited capability to cross the blood–brain barrier. Immune signalling through cytokines produced by the microbiota can also affect brain areas, such as the hypothalamus and circumventricular organs, activating the core stress system, and releasing cortisol. Cortisol will later affect tryptophan metabolism. The vagus nerve has also been involved in many brain-related, microbiota-mediated effects, as has been shown by vagotomy in animal studies, but the detailed mechanism is still unknown. Depression, anxiety disorders, autism, and even ADHD have been linked to microbiota dysbiosis [23,24].

### 3.4. Cardiometabolic Risk

Elevated blood pressure in children is associated with long exposure to PM, especially PM2.5 and NO2. Elevated blood pressure in children is a risk for hypertension and metabolic syndrome later in life. Evidence demonstrates that the effect of air pollution on cardiovascular risk is related to systemic inflammation and immunological responses. The direct contact of pollutants with upper and lower epithelial cells will generate airway oxidative stress and induce local airway inflammation, opening the epithelial’s tight junction barrier, and letting the pollutants enter the bloodstream. There is also a spillover of local lung immune response to the system. This systemic inflammation causes endothelial injury and increases platelet activation, which encourages atherosclerosis and thrombosis. This systemic inflammation also alters the autonomic nervous system and the autorhythmic cells in sinoatrial node and causes cardiac arrhythmias. The combination of all of this increases the risk of cardiovascular diseases, including stroke and myocardial infarction [9,25].

Oxidative stress and systemic inflammation also impair lipoprotein functionality, alter adipocyte behaviour, and disrupt the endocrine process (such as by increasing insulin resistance), and therefore encourage obesity, diabetes mellitus, and dyslipidaemia. All three metabolic risks are strongly associated with cardiovascular risk in the future [9,26].

PM2.5 is the most well-known culprit although PM10 and NO2 also have roles. Coming from ambient or household air pollution and also tobacco smoke, PM exposure has been associated with hypertension, obesity, insulin resistance, dyslipidaemia, cardiac arrhythmia, stroke, and atherosclerosis. Polycyclic aromatic hydrocarbons, however, are the pollutants most strongly associated with obesity [9,26].

Gut microbiota dysbiosis, which can be affected by air pollution, may also increase the risk of cardiometabolic disease in the future. Gut microbiota has been linked with obesity, for example, being one of the risk factors for cardiometabolic diseases. Some of the gut microbiota can hydrolyse and ferment a wide variety of complex polysaccharides that are otherwise indigestible by humans because they lack the specific enzymes. The digestion process will increase the body fat percentage from this energy consumption [27]. The increased plasma concentration of bacterial lipopolysaccharides (LPS), linked with the increased gut permeability caused by the disturbed tight junction, can increase the systemic inflammation linked with obesity and insulin resistance. LPS activates toll-like receptors (TLR), which in turn increase the production of diabetogenic, pro-inflammatory cytokines. This process is called “metabolic endotoxemia” [26]. On the other hand, some species have shown their protection against obesity and insulin resistance by inhibiting pro-inflammatory cytokines, upregulating the expression of tight junction genes, reducing gut permeability and, therefore, decreasing the level of LPS in the plasma. Some species can directly influence glucose homeostasis in metabolic organs, such as the liver, muscles, and fat by increasing the expression of various antidiabetic enzymes. Some of the microbiota, through its metabolism of indigestible complex polysaccharides into short-chain fatty acids (SCFA), such as butyrates, modulate gut hormones and enzymes, improving insulin resistance [28]. While obesity and insulin resistance have been linked to the shifting of the microbial community structure from obligate to facultative anaerobes, no clear pattern has emerged regarding the species [29]. Diversity remains the key in increasing the metabolic capacities of gut microbiota and, therefore, preventing obesity and glucose intolerance-related microbiota dysbiosis [27].

SCFA, specifically propionate, produced by gut microbiota, has also been involved in reducing arterial blood pressure by modulating olfactory receptor 78, which mediates renin secretion, and G protein receptor 41, which regulates the smooth muscle cells of small resistance vessels. Atherosclerotic plaques have also been found to contain bacterial DNA present in the gut of the same individuals. Moreover, trimethylamine N-oxide (TMAO), the hepatic oxidation product of gut-microbiota metabolic waste, trimethylamine (TMA), has been linked with the formation of macrophage foam cells and enhanced atherosclerotic plaque although the direct, causal pathway is still obscured [26,30].

### 3.5. Theoretical Framework

The experts discussed the general impact of air pollution on children’s health. The term “air pollution” here includes indoor and ambient air pollution, as pointed out in the red box. High to moderate levels of evidence support the link between air pollution and the pathway to increase the risk of impaired neurodevelopment, arterial stiffness, oxidative stress, and inflammation, and also the pathway to induce asthma and acute lower respiratory infection. Nevertheless, the pathway from air pollution to metabolic syndrome is still vague. As the gut microbiota has been observed to have a significant link with the neurodevelopmental, immune, and cardiometabolic systems, it is hypothesized that restoring the gut microbiota balance and diversity may disrupt the effect of air pollution on children’s health. However, most existing studies on the gut microbiota have been animal studies and human observations, and no study has been conducted on children specifically. The link between air pollution and gut microbiota also requires further research. Therefore, while hampering the effects of air pollution on children’s health through the manipulation of gut microbiota sounds promising, it is still in a very early, albeit exciting, stage. Figure 3 summarizes the results of the meetings of the experts.

## 4. Conclusions

The evidence of the relationship between air pollution, children’s health, and the protective aspect of the microbiota balance and gut permeability is still limited; yet, the experts agree that these relationships exist to some extent. Air pollution may affect children’s health directly through the neurodevelopmental, immune, and cardiometabolic pathways. However, the available evidence is still insufficient to conclude the relationship between air pollution and the gut microbiota. It is important to gather everything in order to understand the current existing evidence, and the gap needed to be filled. Further research is necessary to explore the pathway in detail and to consider the use of probiotics—live microorganisms that may confer health benefits to the host in adequate doses—to influence health through various pathways [31].

## Figures and Tables

**Figure 1 children-09-00765-f001:**
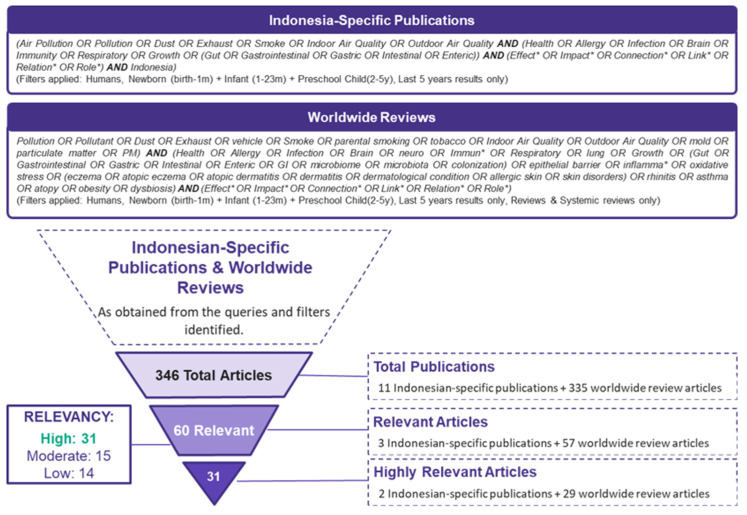
Results of the literature search.

**Figure 2 children-09-00765-f002:**
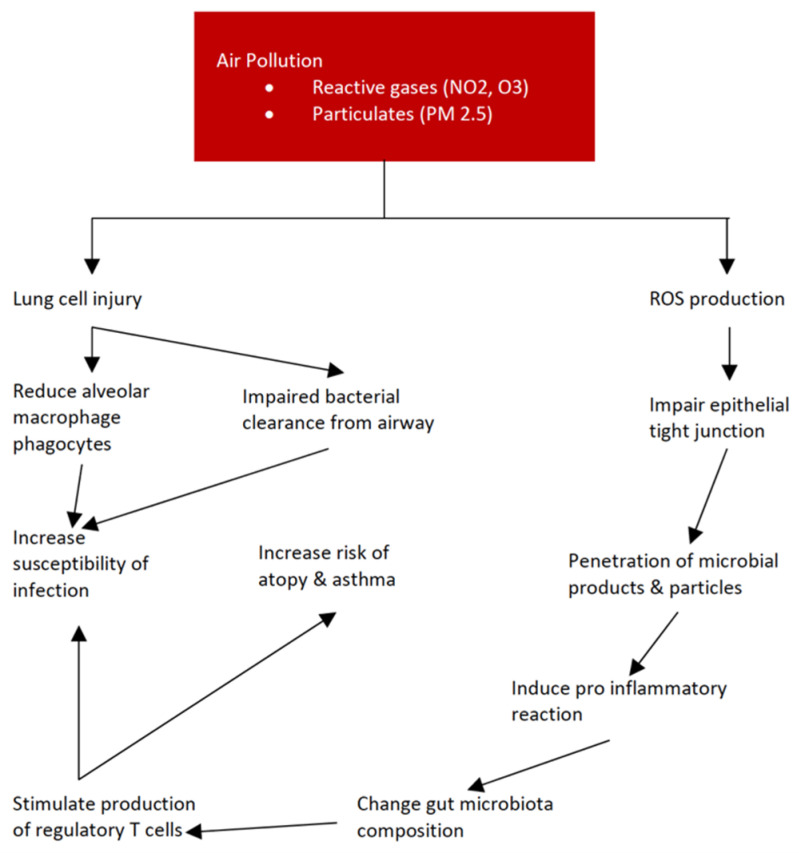
Pathway of the effect of air pollution on immune response (Note: ROS = reactive oxygen species).

**Figure 3 children-09-00765-f003:**
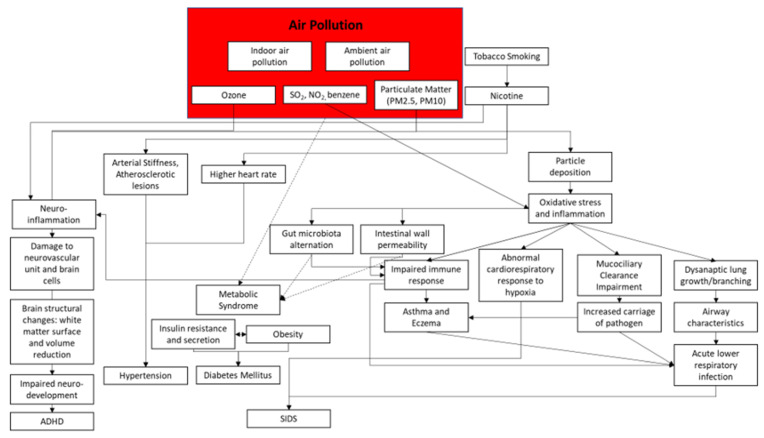
Theoretical framework of the impact of air pollution. (Note: 
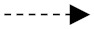
 = weak evidence; 
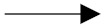
 = strong or moderate evidence; SIDS = sudden infant death syndrome; ADHD = attention deficit hyperactivity disorder; PM = particulate matter).

## Data Availability

Not applicable.

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
