# Peer review of "The Impact of Air Pollution on Gut Microbiota and Children’s Health: An Expert Consensus"

_children, 2022, doi:10.3390/children9060765_

Round 1
Reviewer 1 Report
Thank you for the opportunity to review this manuscript titled, 'Air Pollution Impact to Child's Health in Indonesia; An Expert Consensus'. This review paper primarily focuses on the role air pollution plays on the gut microbiota in young children. This manuscript can only be published provided the following comments and suggestions are adequately addressed.
1) Firstly, the title of the paper is a bit misleading. The authors talk about Indonesia but they have not included any study etc. from Indonesia regarding this topic. If there are no studies, then remove Indonesia from the title.
2) The author should mention gut microbiota in their title as well.
3) Line 2 - In the title, it should be Children's Health and not Child Health
4) Line 57: Define DALYS. Please use the full form of this abbreviation
5) Line 60: Define Dysbiosis as not many people might be aware of this term
6) Line 72: What are the Indonesian air quality standards.
7) Lines 84-84: Please define all these terms: Eubiosis, Dysbiosis, Pro- Pre-Syn-Biotic
8) Could you provide more information on the theoretical framework.
9) Line 151 and 174: Please define ROS and ADHD (Please use the full form and not just abbreviations)
10) Lines 269-275: This is too big a sentence. Divide the content into two or three sentences.
11) Finally how does your conclusion focus on this issue in context of Indonesia?
Author Response
Dear Reviewers,
Thank you for the comments and builds on our article, with running tittle: Air Pollution Impact to Children's Health in Indonesia; An Expert Consensus.
Herewith we resubmit the article for your further review and consideration.
Several revisions on this article consist of additional paragraph in the result and adjusted the conclusion as your feedback.
Best regards,
Dr. dr. Ray Basrowi., MKK
Feedback on the Comments and Suggestions for Authors
Thank you for the opportunity to review this manuscript titled, 'Air Pollution Impact to Child's Health in Indonesia; An Expert Consensus'. This review paper primarily focuses on the role air pollution plays on the gut microbiota in young children. This manuscript can only be published provided the following comments and suggestions are adequately addressed.
- Firstly, the title of the paper is a bit misleading. The authors talk about Indonesia but they have not included any study etc. from Indonesia regarding this topic. If there are no studies, then remove Indonesia from the title.
- The author should mention gut microbiota in their title as well.
- Line 2 - In the title, it should be Children's Health and not Child Health
Based on feedback point 1, 2 and 3, the title has been revised as follow:
The Impact of Air Pollution on Gut Microbiota and Children’s Health: an Expert Consensus
- Line 57: Define DALYS. Please use the full form of this abbreviation
Full term of DALYs has been added:
In fact, 28.9% of the disability adjusted life years (DALYs)
- Line 60: Define Dysbiosis as not many people might be aware of this term
Definition of dysbiosis has been added:
the imbalance condition in the gut microbiota community, or so called dysbiosis
- Line 72: What are the Indonesian air quality standards.
The information on Indonesian air quality standards has been added:
the majority of the average annual PM2.5 exceeded the Indonesian annual ambient air quality standard (15 µg m-3),
- Lines 84-84: Please define all these terms: Eubiosis, Dysbiosis, Pro- Pre-Syn-Biotic
The sentence has been revised:
By understanding how the balance of microbiota community can affect children’s health, it is expected that some form of nurture, such as nutritional supplements can interact and heal the damage caused by air pollution
- Could you provide more information on the theoretical framework.
A paragraph has been added to give more information on the theoretical framework (line 282 – 301)
The experts discussed the general impact on air pollution towards children’s health. The term air pollution here includes indoor and ambient air pollution, as pointed out in the red box. High to moderate level of evidence supported the link between air pollution and the pathway to increase the risk of impaired neurodevelopment, arterial stiffness, oxidative stress and inflammation, and also pathway to induce asthma and acute lower respiratory infection. Nevertheless, the pathway from air pollution to metabolic syndrome was still vague. As gut microbiota has been observed to have a significant link with neurodevelopmental, immune, and cardiometabolic system, it is hypothesized that restoring gut microbiota balance and diversity may disrupt the effect of air pollution on child health. However, most existed studies on gut microbiota has been animal studies and human observation, and no study has been conducted on children, specifically. The link between air pollution and gut microbiota also requires further research. Therefore, while hampering the air pollution effect on child health through manipulation of gut microbiota sound promising, it is still in a very early, albeit exciting, stage. Figure 2 below summarized the result of experts meeting
- Line 151 and 174: Please define ROS and ADHD (Please use the full form and not just abbreviations)
Definition of ROS has been added (line 172)
Both encourage reactive oxygen species (ROS) production, affect epithelial tight junctions, allowing penetration of microbial products and the air pollutant particles to penetrate deeper to the gut wall
Definition of ADHD has been added (line 201)
Behavioural development is significantly affected by air pollution; one example being that the exposure to NO2 and PM2.5 increased the risk for attention deficit hyperactivity disorder (ADHD
- Lines 269-275: This is too big a sentence. Divide the content into two or three sentences.
The sentence as well as the overall conclusion has been revised (Line 303 – 312):
The evidence on the relationship between air pollution, child’s health, and the protective aspect of microbiota balance and gut permeability is still limited, yet the experts agreed that these relationships exist to some extent. Air pollution may affect children’s health directly through neurodevelopmental, immune, and cardiometabolic pathway. But the available evidences were still insufficient to conclude the relationship of air pollution and gut microbiota. It is important to gather everything in order to understand the current existing evidence and the gap needed to be filled. Further researches are necessary to dig the pathway in details and to consider the use of probiotics - live microorganisms that may confer health benefits to the host in the adequate dose—to influence health through various pathways
- Finally how does your conclusion focus on this issue in context of Indonesia?
The conclusion has been revised to adjust with the change of the title

Reviewer 2 Report
Overall: I suspect that a non-native English speaker composed this work as there are significant grammatical and tense issues making understanding the paper very difficult. I would highly recommend working with an English speaking writer to help put sentences in order.
Topic is very interesting and very important. I would be interested in re-reading this manuscript once it has been revised by an English speaker.
I had no idea what the purpose of the paper was by the abstract or introduction. I thought it was a description of the meeting, not a literature review.
Introduction:
- 1st paragraph and 3rd paragraph have some of the same sentences
- All abbreviations need to be defined i.e. WHO needs to first be spelled out World Health Organization
- Many sentences do not have citations.
- “As has been, respiratory in- 54 fection is the second cause of the combination of mortality and morbidity linked with air 55 pollution after neonatal disorders”- respiratory infections are the number one cause of morbidity and mortality in children under five years of age outside of the neonatal time frame (1st two months).
- Writing does not really follow an order making it very hard to read as subjects switch back and forth
Results:
- Subsection (i.e. Allergy and Immunology etc) needs more of an in-depth analysis. These descriptions are very cursory and need graphics and more of a description.
References: Typically literature reviews involve hundreds of citations. I am a bit concerned the depth of this review may not be great enough with such as small number of citations.
Author Response
Dear Reviewers,
Thank you for the comments and builds on our article, with running tittle: Air Pollution Impact to Children's Health in Indonesia; An Expert Consensus.
Herewith we resubmit the article for your further review and consideration.
Several revisions on this article consist of additional paragraph in the result and adjusted the conclusion as your feedback.
Best regards,
Dr. dr. Ray Basrowi., MKK
Feedbacks from the Comments and Suggestions for Authors
Overall: I suspect that a non-native English speaker composed this work as there are significant grammatical and tense issues making understanding the paper very difficult. I would highly recommend working with an English speaking writer to help put sentences in order.
Topic is very interesting and very important. I would be interested in re-reading this manuscript once it has been revised by an English speaker.
I had no idea what the purpose of the paper was by the abstract or introduction. I thought it was a description of the meeting, not a literature review.
Introduction:
- 1st paragraph and 3rd paragraph have some of the same sentences
Similar sentences in 1st paragraph has been revised and the sentences from 3rd paragraph has been deleted:
However, this exposure towards pollutants uniquely affects children because of a combination of behavioural, environmental, and physiological factors, thus make children more vulnerable to air pollution. Children have longer life expectancy, and therefore, longer exposure, and they breathe faster than adults, taking more pollutants through air. Children also live closer to the ground and may spend more time outside, exposing them to higher level of air pollution. Physiologically, their bodies are rapidly changing and maturing, and therefore, more vulnerable towards damages from the pollutants. Moreover, their powerlessness to change their environments is the reason that the adults must be responsible to protect them from the threat of unsafe air [2].
- All abbreviations need to be defined i.e. WHO needs to first be spelled out World Health Organization
All abbreviation has been spelled as full when it is firstly being mentioned
- Many sentences do not have citations.
Citations has been added
- “As has been, respiratory in- 54 fection is the second cause of the combination of mortality and morbidity linked with air 55 pollution after neonatal disorders”- respiratory infections are the number one cause of morbidity and mortality in children under five years of age outside of the neonatal time frame (1st two months).
Sentence has been revised (line 56-57)
As has been, air pollution was responsible for mortality and morbidity due to respiratory infection among under-five children.
- Writing does not really follow an order making it very hard to read as subjects switch back and forth
Sentences has been reviewed and revised
Results:
- Subsection (i.e. Allergy and Immunology etc) needs more of an in-depth analysis. These descriptions are very cursory and need graphics and more of a description.
The subsection has been reviewed and in-depth with more analysis (i.e Line 141 – 193 for allergy and immunology)
References:
Typically literature reviews involve hundreds of citations. I am a bit concerned the depth of this review may not be great enough with such as small number of citations.
The reason/flow of references selection has been explained (line 127 – 140)

Round 2
Reviewer 1 Report
Thank you for submitting the revised version of the manuscript. The authors have substantially revised the manuscript and it can now be accepted for publication. I do not have any further comments.